# The Impact of the Dietary Intake of Vitamin B12, Folic Acid, and Vitamin D3 on Homocysteine Levels and the Health-Related Quality of Life of Levodopa-Treated Patients with Parkinson’s Disease—A Pilot Study in Romania

**DOI:** 10.3390/diagnostics14151609

**Published:** 2024-07-26

**Authors:** Adina Turcu-Stiolica, Mihaela-Simona Naidin, Steliana Halmagean, Ana Maria Ionescu, Ionica Pirici

**Affiliations:** 1Pharmaceutical Management and Marketing, Faculty of Pharmacy, University of Medicine and Pharmacy of Craiova, 200349 Craiova, Romania; adina.turcu@umfcv.ro; 2Neuro Therapy Clinic, 300425 Timisoara, Romania; steliana.halmagean@neurotherapy.ro; 3Department of Neurology, Ovidius University, 900123 Constanta, Romania; ana.maria.ionescu@univ-ovidius.ro; 4Department of Human Anatomy, University of Medicine and Pharmacy of Craiova, 200349 Craiova, Romania; ionica.pirici@umfcv.ro

**Keywords:** Parkinson’s disease, homocysteine, folic acid, vitamin D, vitamin B12, quality of life

## Abstract

Background and Objectives: Previous studies have shown that the levodopa treatment of Parkinson’s disease (PD) elevates circulating homocysteine levels, which are associated with an increased risk of cardiovascular and neurological disorders, or thrombosis. The present trial aimed to examine whether the intake of vitamin B12, folic acid, and vitamin D3 supplements improved homocysteine level and quality of life (QoL). Materials and Methods: An interventional prospective trial was conducted in multiple centers across Romania. Participants with clinically established PD taking at least 300 mg/day of levodopa for more than 1 year received a daily tablet of a supplement containing 800 UI of vitamin D3, 1000 µg of folic acid, and 15 µg of vitamin B12. They were followed for 6 months and their serum homocysteine, vitamin B12, vitamin D, and QoL scores were measured at baseline and at 6 months of treatment. QoL was measured using a 15D questionnaire, which assesses mobility, vision, hearing, breathing, sleeping, eating, speech, excretion, usual activities, mental function, discomfort and symptoms, depression, distress, vitality, and sexual activity. Results: Twenty-four PD patients with a mean age of 71 ± 5.04 years (54.2% male and 45.8% female) finished the study. After the intervention, the mean score of speech, mental function, discomfort and symptoms, depression, and QoL significantly increased (*p* < 0.05 for all). Also, the serum homocysteine and vitamin D were significantly enhanced (*p* < 0.0001 and *p* = 0.025, respectively). Changes in vitamin B12 were not statistically significant at 6 months of treatment (*p* = 0.996). No gender differences were found among the changes that we have demonstrated for homocysteine, vitamin B12, vitamin D, and QoL levels (*p* < 0.05 for all). Conclusions: The findings of this study showed that the dietary intake of vitamin B12, folic acid, and vitamin D3 remarkably decreased the dimensions of homocysteine and finally increased the total score of QoL in PD patients. We have successfully captured the potential benefits of the supplementation regimen over time and provided insights into the broader implications for managing PD with a focus on nutritional support.

## 1. Introduction

Parkinson’s disease (PD) is a neurodegenerative disease, characterized by dopamine deficiencies, which leads to motor symptoms, including bradykinesia, resting tremor, rigidity, and non-motor symptoms such as sleep disorders, anosmia, constipation, and depression [1]. PD is a progressive disorder, which means that its symptoms worsen over time [2,3]. The exact cause of Parkinson’s disease is not fully understood, but it is believed to involve a combination of genetic, environmental, and neurochemical factors [3]. In Parkinson’s, there is a progressive loss of dopaminergic neurons in a part of the brain called the substantia nigra [4]. This leads to a decrease in dopamine levels, a neurotransmitter crucial for motor control and other functions [5].

During the last 3 decades, the worldwide incidence of PD has doubled in number and is projected to double again by the year 2040 [6,7]. While there is no cure for Parkinson’s disease, there are several treatment options available to manage its symptoms and improve quality of life. Medications with dopamine replacement therapy, such as levodopa, dopamine agonists, and monoamine oxidase-B inhibitors, can help alleviate motor symptoms [8]. Other groups of drugs, such as monoamine oxidase B inhibitors, catechol-O-methyltransferase inhibitors, and amantadine, are also seen as background therapy [9]. Physical therapy, occupational therapy, and speech therapy can also be beneficial [10].

The results of recent years have shown that oxidative stress may play a critical role in the onset and the progression of Parkinson’s disease (PD) [11]. Neurons are more vulnerable to the deterioration of folate deficiency (FD) and vitamin B12 (Vit B12), as well as the increase in homocysteine (Hcy); the underlying mechanisms may increase oxidative stress and low methylation [12]. In parallel, Hcy may play a role in the onset or progression of PD through gene defects and apoptosis [13]. Hcy has been found to exhibit multiple neurotoxic effects and it is rational to infer that high dietary intakes of B vitamins may lower the risk of PD by decreasing plasma Hcy levels [14]. Vit B12 is a vitamin with a role in DNA synthesis and the decrease in or absence of this vitamin determines pathologies from the hematological, neurological, and psychiatric spectrum. The level of this vitamin decreases with advancing age and this decline can be accentuated using gastric acid-blocking agents.

FA (folic acid) and Vit B12 are necessary cofactors in Hcy metabolism [13] and may have an underlying association with PD onset or progression [15]. The serum levels of Vit B12 and FA have an inverse relationship with the plasma level of Hcy. Hcy, Vit B12, and FA levels are associated with each other. The major causes of hyperhomocysteinemia are deficiencies in FA and Vit B12, which are necessary for Hcy metabolism [16]. In addition, deficiencies in FA and Vit B12 are associated with neural degeneration [17]. If low FA and Vit B12 levels are associated with greater disability, treatment studies using FA and Vit B12 supplements should be considered.

Apart from variations in therapeutic response, the persistent use of levodopa during advanced stages of the disease may lead to an accentuation of adverse side effects. Levodopa-induced dyskinesia, a neurological disorder marked by involuntary choreatic and dystonic movements of the extremities, commonly manifests in patients undergoing prolonged levodopa therapy [18]. In recent times, there has been an increased focus on the connection between nutrition and medicinal treatment, as certain dietary patterns have been found to enhance the efficacy of drugs. It is not unexpected that nutritional interventions could play a significant role in optimizing levodopa therapy [18]. FA and Vit B12 may be included in the supplementation regimen for the defense against neurodegeneration in PD [19].

The role of vitamin D as a neuroprotective factor has been widely described, but the connection of this vitamin with PD is not fully explained. In fact, this relationship was appreciated by some authors through a negative correlation, namely by the fact that the concentration of vitamin D does not correlate with PD progression assessed using the UPDRS score [20,21], and other authors published studies that showed an inverse relationship between increased values of UPDRS scores and Hoehn and Yahr (HY) staging and low vitamin D levels [22,23].

It has been proven that the levodopa treatment of PD elevates circulating Hcy levels [8]. Hyperhomocysteinemia in PD patients may contribute to depression and Alzheimer’s disease, being an independent factor for PD progression [24,25,26,27]. The mechanism of hyperhomocysteinemia is presented in Figure 1.

In PD, Hcy levels become elevated due to multiple disruptions in its metabolism. These disruptions include the impaired remethylation of Hcy to methionine due to the reduced activity of methionine synthase (MS) and methylenetetrahydrofolate reductase (MTHFR), often exacerbated by lower levels of vitamin B12 and 5-MTHF [28]. Additionally, the trans-sulfuration pathway, which converts Hcy to cysteine via cystathionine-β-synthase (CBS) and requires vitamin B6, is also compromised. PD increases the demand for S-adenosylmethionine (SAM), resulting in higher Hcy production [29]. Furthermore, disruptions in folate metabolism, including the impaired conversion of folate to tetrahydrofolate (THF) and subsequently to 5-MTHF, affect Hcy remethylation. These metabolic abnormalities highlight the importance of sufficient levels of vitamins B6, Vit B12, and FA to support normal Hcy metabolism and address metabolic complications in PD [30].

Muller et al. demonstrated that levodopa induces vitamin deficiency in PD, resulting in falls and reduced quality of life [31]. Along with physical problems, the mixture of anxiety and depression in PD patients reduces their health-related quality of life (HRQoL) [32].

Based on multiple studies indicating that levodopa induces vitamin deficiency and reduces the quality of life in patients with PD, we aimed to analyze the changes in levels of Hcy, Vit B12, and Vit D, as well as the changes in the HRQoL among patients with PD following a 6-month pilot study of a treatment regimen that included supplements containing Vit B12 (5 µg), FA (1000 µg), and Vit D3 (800 IU). This study involved the regular monitoring and assessment of the patients’ vitamin levels and HRQoL metrics at the beginning and end of the study. This approach allowed us to capture the potential benefits of the supplementation regimen over time and provided insights into the broader implications for managing PD with a focus on nutritional support.

## 2. Materials and Methods

### 2.1. Study Design

This is a pilot interventional study where the PD patients were followed prospectively after 6 months of treatment with supplements of Parkovit@, containing Vit B12 (5 µg), FA (1000 µg), and Vit D3 (800 IU). This study was conducted in five Neurology Centers across Romania. All of the included patients had a confirmed diagnosis of PD by a neurologist based on the Hoehn and Yahr Scale [33]. The inclusion criteria were the following: (1) over 18 years old, (2) patients with PD taking at least 300 mg/day of levodopa for more than 1 year, and (3) patients who signed informed consent to be included in the study and were capable of reporting on their HRQoL by answering the 15D questionnaire. Patients following special diets (calorie restrictions, low protein, supplements such as vitamins B1/B6/B12/D/E, omega-3, etc.) were excluded from the study. Also, patients with gastroenterological conditions (e.g., malabsorption, inflammatory bowel disease) were not eligible. Demographic characteristics (age and gender), PD stage, comorbidities, and medication were obtained at baseline using a general questionnaire. Serum Hcy, Vit B12, Vit D, and HRQoL scores were measured at baseline and after 6 months of treatment.

### 2.2. Ethical Considerations

We obtained ethical approval for our research project from the University of Medicine and Pharmacy of Craiova Ethics Commission (no. 49/10 February 2023). All the collected data were stored according to the ethical guidelines of medical research.

### 2.3. Data Collection

Patients were required to complete the self-reported questionnaire focused on quality of life at baseline and after the 6-month period of treatment. HRQoL was evaluated using the 15D instrument, a validated generic questionnaire that assesses 15 dimensions: mobility, vision, hearing, breathing, sleeping, eating, speech, excretion, usual activities, mental function, discomfort and symptoms, depression, distress, vitality, and sexual activity. The overall HRQoL was quantified using a 15D score, ranging from 0 (representing health states regarded as bad, with the lowest score being dead) to 1 (representing full health). The 15D instrument, translated for the Romanian population, was employed in this study [34]. A significant clinical change in HRQoL was defined as a minimum important difference of 0.015 in the 15D score from baseline [35]. The categories for change were defined as follows: “much better” (difference greater than 0.035), “slightly better” (difference between 0.015 and 0.035), “much the same (no change)” (difference between −0.015 and 0.015), “slightly worse” (difference between −0.035 and −0.015), and “much worse” (difference less than −0.035).

Hcy, Vit B12, and Vit D were measured in a clinical laboratory, with the following reference ranges: for Hcy [6.6–17.8 μmol/L]), normal levels of Vit B12 [211–911 pg/mL]), and the normal range for Vit D [30–100 ng/mL]. Hcy levels were assessed in the laboratory using a fluorescence enzyme immunoassay [36]. Vit D levels were assessed in the laboratory by measuring the concentration of 25-hydroxyvitamin D in the blood, as it is the major circulating form of Vit D and is considered the best indicator of Vit D status [37]. The technique used for measuring Vit B12 levels was the immunoassay, where the signal was measured with a chemiluminescent analyzer [38].

### 2.4. Data Analysis

The normal distribution of continuous variables was confirmed using the Kolmogorov–Smirnov test. A two-tailed Wilcoxon matched-pairs signed-rank test was used to detect differences in serum Hcy, Vit B12, Vit D, and HRQoL scores between baseline and after 6 months of treatment. A two-tailed Mann–Whitney U Test was used to detect differences in serum Hcy, Vit B12, Vit D, and HRQoL scores between male and female patients. A *p*-value of less than 0.05 was considered the level of significance. All statistical analyses were conducted using Python (v3.11, Python Software Foundation). To visualize the distribution and difference before and after the Parkovit@ treatment, violin plots were generated using Seaborn’s violin plots function. A Seaborn Heatmap was used to quickly visualize the correlations between the improvement or worsening in Hcy, Vit B12, Vit D, and HRQoL and its dimensions (mobility, sleeping, eating, speech, excretion, usual activities, mental function, discomfort and symptoms, and depression) levels (colors range from bright blue for a strong positive correlation to bright red for a strong negative correlation). The power analysis of our results was performed using G*Power 3.1.9.7; it was two-tailed, post hoc, assuming an alpha level of 0.05.

## 3. Results

Thirty-two PD patients finished the study, but only twenty-four with a mean age of 71 ± 5.04 years (54.2% male and 45.8% female) completed all of the data. The characteristics of the included patients are presented in Table 1. A number of eight patients from the thirty-two patients did not complete the HRQoL questionnaire, as displayed in Figure 2. Thirty-four patients were withdrawn due to their lack of willingness to continue the study (dropout rate of 44%). No side effects were reported.

Using stringent criteria, 17 (71%) of the patients had hyperhomocysteinemia at baseline, and only 4 (17%) patients had hyperhomocysteinemia after the 6-month treatment. Regarding the Vit D levels, 22 patients (92%) had an insufficient level of Vit D at baseline, and 17 patients (71%) had insufficient levels after the 6-month treatment.

As shown in Table 2, compared to baseline, after 6 months of treatment, the Hcy levels significantly decreased (*p* < 0.0001), while Vit D levels significantly increased (*p* = 0.025). After 6 months of treatment, the Vit B12 levels did not change significantly (*p* > 0.05). Only 8% of the cohort was B12-deficient at baseline and this percentage remained unchanged after the 6-month treatment.

According to the responses to the HRQoL questions, the patients declared a statistically significant improvement in quality of life (0.75 ± 0.11, *p*-value = 0.0246). As shown in Figure 3 and Table 3, we observed no differences in mobility, sleeping, eating, excretion, or usual activities (*p* > 0.05 for all).

The improvement in QoL was correlated with an increase in vitamin B12 levels (Spearman’s rho = 0.408, *p* = 0.048). No difference in improvement of QoL was observed between the men and women (Spearman’s rho = −0.006, *p* = 0.978), as shown in Figure 4.

Furthermore, we investigated whether there were differences in these outcomes depending on gender. By analyzing the data separately for male and female patients, we aimed to identify any gender-specific responses to the supplementation regimen. No gender differences were observed, as shown in Table 4.

## 4. Discussion

The benefit of FA, Vit D3, and Vit B12 supplementation in Parkinsonian patients treated with levodopa to prevent hyperhomocysteinemia has been established, but no studies have previously measured the HRQoL.

One objective of our follow-up study was to determine the plasma Hcy levels in levodopa PD patients after 6 months of taking one daily tablet of Parkovit@. We demonstrated that this treatment decreases Hcy levels, similar to the findings of Muller et al. [32].

Our current research demonstrates that dietary intervention with one daily tablet of Parkovit@ (containing 800UI of Vit D3, 1000 µg of FA, and 15 µg of Vit B12) can improve the quality of life in patients with PD.

Additionally, a significant positive correlation was determined between the improvement of QoL and the increase in Vit B12 levels after 6 months of treatment. Among all vitamins, Vit B12 appeared to have the most robust link with PD [39]. Zalyalova et al. demonstrated that the use of high-dose (1000 µg) oral Vit B12 has comparable clinical efficacy with the parenteral form in PD [40].

Depression among individuals with Parkinson’s disease (PD) is primarily characterized by diminished pleasure and enjoyment, often accompanied by dissatisfaction and reduced appetite. However, feelings of guilt, self-loathing, and decreased libido are less commonly reported, despite sharing fundamental diagnostic criteria with depression and anxiety disorders in the broader population [41]. We tested the hypothesis that the QoL, inclusive of depression and mental function, of levodopa-treated PD patients was enhanced after Parkovit@ treatment, and our results could be related to the results of Triantafyllou et al., who associated lower serum folate with depression in levodopa-treated PD patients [42]. In our study, depression and mental function were improved after the 6-month treatment period. Discomfort and symptoms were also improved and could lead to a virtuous cycle.

Constipation is one of the common non-motor symptoms of Parkinson’s disease that significantly impacts patient QoL [43]. Our findings showed that excretion function, as measured using the 15D QoL questionnaire, improved significantly after the 6 months of supplementation with Parkovit@.

PD extends beyond its motor symptoms, encompassing various non-motor manifestations that hold significance for distinct reasons [44]. These non-motor symptoms are not only prevalent but also significantly impairing, underscoring the importance of early detection and effective management [45]. Certain non-motor symptoms, such as olfactory disturbances, constipation, or sleep disturbances, may manifest before motor symptoms, serving as potential markers for the prodromal or preclinical stages of PD [46]. Conversely, other symptoms like dementia and psychosis tend to emerge more commonly in the disease’s later stages and can pose challenges in treatment. Understanding the intricate progression of non-motor symptoms and their interplay with motor changes over time is imperative for comprehensive management [47].

Another QoL indicator, sleep, is also closely related to the management of PD treatment. Loddo et al. reviewed the principal therapeutic options for the most common sleep disorders in PD patients, noting that while their treatment is challenging, it significantly helps improve the management of PD and the QoL in these patients [48]. Zalyalova et al. highlighted that in long-term treatment with levodopa, especially including intraduodenal administration, REM sleep behavioral disorders are associated with Vit B12 deficiency [40].

It is imperative to prioritize the identification of evolving changes in the QoL among PD patients and the increasing burden experienced by their caregivers over time [49]. Additionally, understanding the factors that may anticipate these alterations is crucial for implementing effective interventions. Conducting well-structured longitudinal studies is essential for this purpose [50]. Access to a population comprising a significant number of patients who have undergone comprehensive and meticulous assessments, devoid of screening biases, holds considerable significance for both cross-sectional examinations and prospective monitoring [51]. This significance is particularly pronounced when studying cohorts affected by neurodegenerative diseases, given the anticipation of diverse complications among these patients that warrant identification and analysis [52,53].

Our study demonstrated statistically significant increases in Vit D levels after 6 months of Parkovit@ treatment. In PD, Vit D deficiency results in decreased levels of dopamine, and an imbalance in Vit D leads to the accumulation of synuclein [47]. Ensuring adequate Vit D levels might offer neuroprotective benefits, potentially improving disease outcomes and slowing progression. However, more research is needed to fully elucidate the mechanisms and to determine the therapeutic potential of Vit D supplementation in PD patients.

As gender differences in disease progression and treatment response are increasingly recognized in PD research [54,55], we investigated whether there was a gender-specific response to the supplementation regimen. No gender differences were found among the changes we have demonstrated for Hcy, Vit B12, Vit D, and HRQoL levels. Despite an extensive literature search, we did not find previous studies that specifically addressed a combination of vitamin supplementation and its impact on both vitamins and HRQoL in PD patients undergoing levodopa treatment. Therefore, our findings represent a novel contribution to the field, highlighting the need for further research to validate and expand upon our results. Future studies should aim to replicate our findings and explore the underlying mechanisms that might explain gender differences in response to vitamin supplementation in the context of PD.

Some limitations of the present study should be mentioned. One limitation is the small number of patients, for which we calculated the sample power. The results yielded an achieved power between 76% and 99.5% for the different analyses. Another limitation is that our results may suffer from a lack of controls for other factors such as the duration of disease and cognitive impairment. Despite these limitations, our study is the first study to assess the impact of the dietary intake of Vit B12, FA, and Vit D3 on quality of life among Romanian patients with PD and further research is needed on a larger sample size.

## 5. Conclusions

In conclusion, additional folic acid, Vit B12, and Vit D3 supplementation with the concomitant lowering of total Hcy levels in levodopa-treated PD patients will hypothetically increase the quality of life in treated PD patients. This study will help in developing more exact doses and protocols for treatment with folate, Vit B6, and Vit B12 in levodopa-treated PD patients.

## Figures and Tables

**Figure 1 diagnostics-14-01609-f001:**
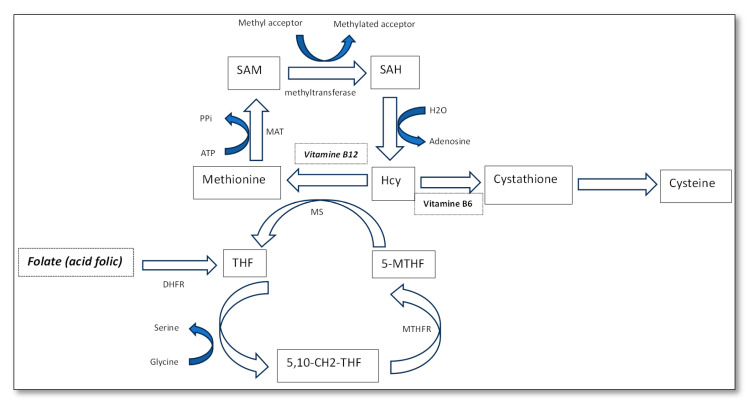
Homocysteine metabolism. 5-MTHF, 5-methyltetrahydrofolate; THF, tetrahydrofolate; 5,10-CH2-THF, 5,10-methylenetetrahydrofolate; MTHFR, 5,10-methylenetetrahydrofolatereductase; MS, methionine synthetase; MAT, methionine adenosyltransferase or S-adenosylmethionine synthetase; DHFR, Dihydrofolate Reductase; S-adenosylmethionine (SAM); S-adenosine homocysteine (SAH).

**Figure 2 diagnostics-14-01609-f002:**
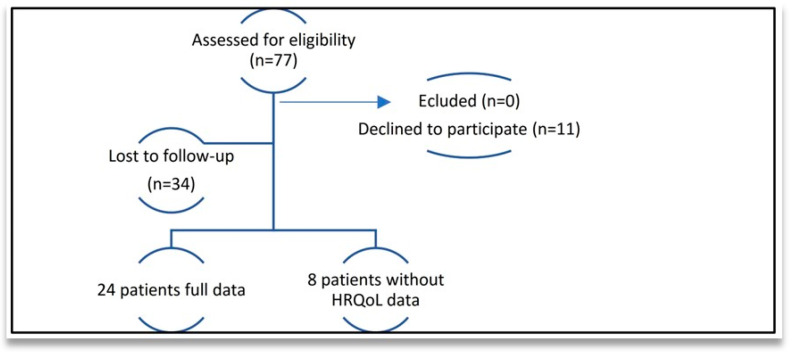
CONSORT flow diagram of study recruitment.

**Figure 3 diagnostics-14-01609-f003:**
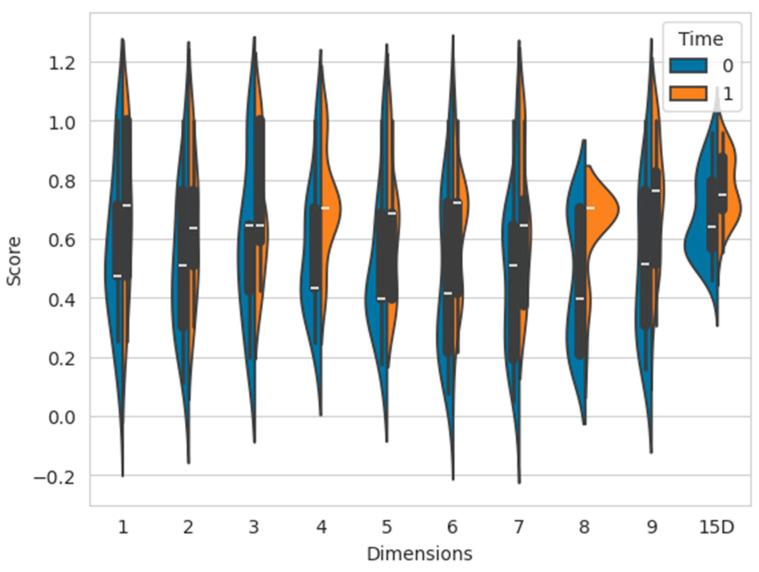
Differences between baseline (time = 0) and after 6 months of treatment (time = 1) among the 15 dimensions of treatment. 1 = mobility; 2 = sleeping; 3 = eating; 4 = speech; 5 = excretion; 6 = usual activities; 7 = mental function; 8 = discomfort and symptoms; 9 = depression; 15D = Health-Related Quality of Life score. Compared to the QoL before treatment, the overall health status is significantly improved, as shown in Table 2.

**Figure 4 diagnostics-14-01609-f004:**
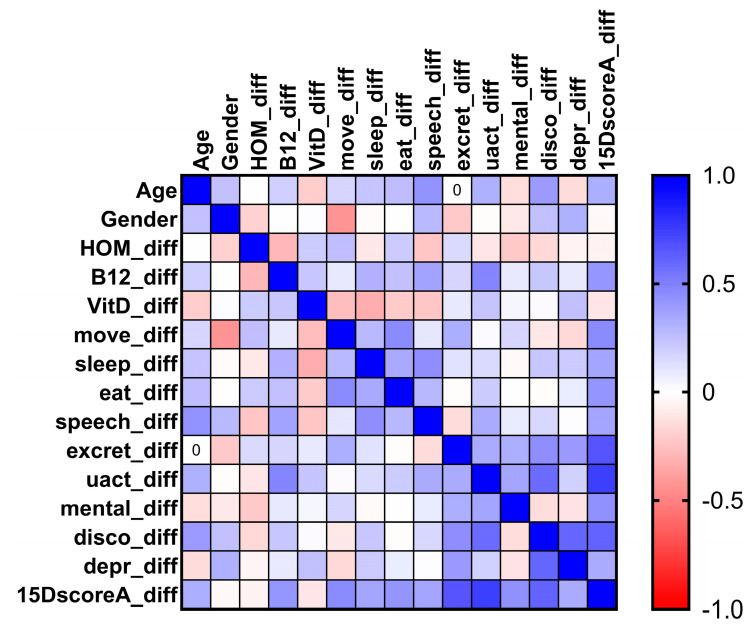
Heatmap of correlations among improvement or worsening of homocysteine (HOM), vitamin B12 (B12), vitamin D (vitD), and HRQoL (15D score) and its dimension (Move = mobility, Sleep = sleeping, Eat = eating, Speech = speech, Excret = excretion, Uact = usual activities, Mental = mental function, Disco = discomfort and symptoms, Depr = depression) levels (colors range from bright blue for a strong positive correlation to bright red for a strong negative correlation).

**Table 1 diagnostics-14-01609-t001:** Characteristics of the patients.

Characteristics	Baseline (*n* = 24)
Age, years	71 ± 5.04
Gender, male	13 (54.2%)
Parkinson’s disease stage	
II H&I	9 (37.5%)
III H&I	9 (37.5%)
IV H&I	6 (25%)
Comorbidities	
Obesity	18 (75%)
Ischemic heart disease	8 (33.3%)
Hypertension	7 (29.2%)
Diabetes type 2	6 (25%)

Continuous variables were reported as means ± standard deviations. Categorical variables are reported as numbers (percentages). H&I, Hoehn and Yahr staging.

**Table 2 diagnostics-14-01609-t002:** Comparison of vitamins between baseline and after 6 months of supplement treatment.

Vitamins Mean ± SD Median (IQR)	Baseline	After 6 Months of Treatment	*p*-Value
Homocysteine	19.98 ± 4.995 19.6 (16.4–23.10)	16.18 ± 3.73 15.9 (13.60–18.28)	<0.0001 ****
Vitamin B12	352.3 ± 119.1 331.0 (265.5–417)	345.8 ± 120.9 309.5 (251.8–439.5)	0.9963
Vitamin D	18.96 ± 7.28 18.4 (14.23–24.16)	22.68 ± 7.65 21.51 (16.75–29.78)	0.025 *

Two-tailed Wilcoxon matched-pairs signed-rank test. *, *p* < 0.05, ****, *p* < 0.0001.

**Table 3 diagnostics-14-01609-t003:** Comparison of the 15D dimensions between baseline and after 6 months of supplement treatment.

Dimension of Quality of Life	Difference Mean ± SD	Statistical Significance (*p*-Value)	Clinical Significance
Mobility	0.11 ± 0.13	0.1444	much better
Sleeping	0.12 ± 0.15	0.0994	much better
Eating	0.11 ± 0.15	0.1109	much better
Speech	0.15 ± 0.13	0.0126 *	much better
Excretion	0.11 ± 0.14	0.0872	much better
Usual activities	0.14 ± 0.18	0.0536	much better
Mental function	0.17 ± 0.17	0.0230 *	much better
Discomfort and symptoms	0.18 ± 0.19	0.0024 **	much better
Depression	0.21 ± 0.15	0.0053 **	much better
15D score	0.09 ± 0.05	0.0246 *	much better

*, *p*-value < 0.05; **, *p*-value < 0.01. “much better” (difference greater than 0.035), “slightly better” (difference between 0.015 and 0.035), “much the same (no change)” (difference between −0.015 and 0.015), “slightly worse” (difference between −0.035 and −0.015), and “much worse” (difference less than −0.035).

**Table 4 diagnostics-14-01609-t004:** Comparison of the changes in the vitamins and HRQoL levels between male and female PD patients.

Outcomes Changes Mean ± SD	Male (n = 13)	Female (n = 11)	*p*-Value
Homocysteine changes	4.65 ± 3.45	3.87 ± 3.05	0.392
Vitamin B12 changes	20.46 ± 102.8	29.82 ± 134.9	1.0
Vitamin D changes	2.44 ± 5.22	4.27 ± 9.7	1.0
Mobility changes	0.17 ± 0.12	0.05 ± 0.11	0.063
Sleeping changes	0.12 ± 0.18	0.12 ± 0.12	0.955
Eating changes	0.11 ± 0.16	0.11 ± 0.13	1.0
Speech changes	0.13 ± 0.13	0.18 ± 0.14	0.228
Excretion changes	0.14 ± 0.14	0.08 ± 0.13	0.361
Usual activities changes	0.14 ± 0.19	0.16 ± 0.18	0.733
Mental changes	0.19 ± 0.18	0.12 ± 0.15	0.392
Discomfort and symptoms changes	0.14 ± 0.17	0.23 ± 0.21	0.277
Depression changes	0.16 ± 0.11	0.27 ± 0.17	0.150
HRQoL changes	0.09 ± 0.06	0.09 ± 0.02	1.0

A two-tailed Mann–Whitney U Test was performed.

## Data Availability

The raw data supporting the conclusions of this article will be made available by the authors on request.

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
