# Peer review of "The Impact of the Dietary Intake of Vitamin B12, Folic Acid, and Vitamin D3 on Homocysteine Levels and the Health-Related Quality of Life of Levodopa-Treated Patients with Parkinson’s Disease—A Pilot Study in Romania"

_diagnostics, 2024, doi:10.3390/diagnostics14151609_

Round 1

Reviewer 1 Report

Comments and Suggestions for Authors

Turcu-Stiolica et al. conducted a prospective trial to assess the impact of vitamin B12, folate, and D3 dietary intake on homocysteine and HRQoL in PD with levodopa therapy in Romania. The study, which involved 24 participants, revealed significant enhancements in serum homocysteine and vitamin D levels (p < 0.0001 and p = 0.025, respectively), while changes in vitamin B12 were not statistically significant at six months (p = 0.996). These findings underscore the potential influence of dietary factors on PD progression. My recommendations are:

The tile is appropriate to the current manuscript.

I recommend that the abstract include more data about the result section. Although the methodology was extensively described, the results should be the center of the abstract.

Keywords are appropriate.

Please explain in Figure 1 the proposed mechanism for abnormalities in homocysteine levels secondary to PD.

Please explain how the laboratory values were assessed; the techniques should be described with a reference.

Had all the variables a normal distribution?

Please provide a complete description of the software used, including the version and city of development (Python).

Please provide a table with the baseline demographic characteristics of the individuals.

Is this a test of a new drug (Parkovit@)? Does this study need a trial registration? How is the approval of this medication in the author’s country? Please provide a link with the regulatory agency's approval for review; it does not need to be included in the manuscript.

Author Response

Dear Reviewer,

We appreciate the time and effort you dedicated to reviewing our work thoroughly. We have diligently addressed each point raised, implementing necessary corrections in our manuscript. We hope the revisions made using the tracked changes feature has significantly improved the quality of our manuscript.

The tile is appropriate to the current manuscript.

I recommend that the abstract include more data about the result section. Although the methodology was extensively described, the results should be the center of the abstract.

Agree. We have accordingly revised the abstract to empasize this point.

This change can be found at the page number 1, and lines 31-37.

Keywords are appropriate.

Please explain in Figure 1 the proposed mechanism for abnormalities in homocysteine levels secondary to PD.

Thank you for your recommendation. We updated the text in the manuscript. This change can be found at the page number 3, and lines 99-109.

Please explain how the laboratory values were assessed; the techniques should be described with a reference.

Thank you for pointing this out. We agree with this comment.

Homocysteine levels were assessed in the laboratory using Fluorescence Enzyme Immunoassay [Paprotny, L.; Wianowska, D.; Izdebska, M.; Celejewska, A.; Szewczak, D.; Solski, J. Analysis of serum homocysteine in the laboratory practice - comparison of the direct chemiluminescence immunoassay and high performance liquid chromatography coupled with fluorescent detection. Biochem Med (Zagreb). 2020, 30(3), 030703. doi: 10.11613/BM.2020.030703.]. Vitamin D leveles were assessed in the laboratory by measuring the concentration of 25-hydroxyvitamin D in the blood as it is the major circulating form of vitamin D and is considered the best indicator of vitamin D status. [Binkley, N.; Sempos, C.T. Vitamin D Standardization Program (VDSP). Standardizing vitamin D assays: the way forward. J Bone Miner Res. 2014, 29(8), 1709-14. doi: 10.1002/jbmr.2252.]. The technique used for measuring Vitamin B12 levels was the immunoassays, where the signal was measured with a chemiluminescent analyser [Rukhsana J, Perrotta PL, Okorodudu AO, Petersen JR, Mohammad AA. Fit-for-purpose evaluation of architect i1000SR immunoassay analyzer. Clin Chim Acta. 2010, 411(11-12), 798-801. doi: 10.1016/j.cca.2010.02.061.].

This change can be found at the page number 4, and lines 170-175.

Had all the variables a normal distribution?

We verified the normal distribution of the variables using the Kolmogorov-Smirnov test. Not all variables exhibited a normal distribution; for example, the distributions of the variables Moving, Eating, Speech, Excretion, Usual Activities, Mental, and Discomfort were not normal. Furthermore, we used non-parametric statistical tests due to the small sample size of the patient cohort.

Please provide a complete description of the software used, including the version and city of development (Python).

We have added the information.

This change can be found at the page number 4, and line 184.

Please provide a table with the baseline demographic characteristics of the individuals.

We have added a table with the baseline characteristics of the individuals. Thank you.

This change can be found at the page number 5, and lines 195-202.

Is this a test of a new drug (Parkovit@)? Does this study need a trial registration? How is the approval of this medication in the author’s country? Please provide a link with the regulatory agency's approval for review; it does not need to be included in the manuscript.

Parkovit@ is not a new drug. It is a supplement containing Vit B12 (5 µg), FA (1000 µg), and Vit D3 (800 IU). Our study is an observational study for which we have obtained the ethical approval from The University of Medicine and Pharmacy of Craiova Ethics Commission (no. 49/10.02.2023).

Reviewer 2 Report

Comments and Suggestions for Authors

In the manuscript submitted for review, the Authors analyzed impact of dietary intake of vitamin B12, folic acid, and vitamin D3 on homocysteine level and the health-related quality of life in levodopa-treated patients with Parkinson’s disease. I find the topic of the manuscript interesting and "up to date" and the whole work is thoughtful. The Authors put a lot of work into preparing this interesting work. A well-written, comprehensive introduction to the issue deserves attention.

My comments:

1. In my opinion, at the end of the introduction there should be a clearly formulated research hypothesis

2. I did not find information whether there were differences depending on gender. Did the Authors take this aspect into account?

Author Response

Dear Reviewer,

We appreciate the time and effort you dedicated to reviewing our work thoroughly. We have diligently addressed each point raised, implementing necessary corrections in our manuscript. We hope the revisions made using the tracked changes feature has significantly improved the quality of our manuscript.

  1. In my opinion, at the end of the introduction there should be a clearly formulated research hypothesis

Thank you for pointing this out. We agree with this comment. We have added more details at the end of the introduction.

This change can be found at the page number 3, and lines 122-130.

  1. I did not find information whether there were differences depending on gender. Did the Authors take this aspect into account?

We appreciate your insights and have incorporated an analysis of gender differences. However, we did not find any studies to compare our results with in the discussion. Therefore, we have included our comments in the discussion section.

This change can be found at the page number 7 (lines 248-254) and the page number 9 (lines 320-330).
